# Brain Plasticity Profiling as a Key Support to Therapeutic Decision-Making in Low-Grade Glioma Oncological Strategies

**DOI:** 10.3390/cancers15143698

**Published:** 2023-07-20

**Authors:** Sam Ng, Hugues Duffau

**Affiliations:** 1Department of Neurosurgery, Gui de Chauliac Hospital, Montpellier University Medical Center, 34295 Montpellier, France; h-duffau@chu-montpellier.fr; 2Institute of Functional Genomics, University of Montpellier, Centre National de le Recherche Scientifique, Institut National de la Santé et de la Recherche Médicale 1191, 34094 Montpellier, France

**Keywords:** brain plasticity, neuroplasticity, low-grade glioma, surgery, awake mapping, quality of life, white matter connectivity, glioma recurrence

## Abstract

**Simple Summary:**

Diffuse low-grade gliomas promote widely distributed mechanisms of cerebral plasticity, which are critical to maintaining brain functions. These neurobiological processes are highly correlated to the success and the risks of therapies, and, reciprocally, current treatment options have a critical effect on the long-term potentiation or inhibition of plasticity. In this narrative review, we discuss current clinical, radiological, and oncological markers that reflect plasticity potentials and limitations in the context of low-grade glioma management, and we further highlight how plasticity interacts with spontaneous and therapeutic factors. Finally, we propose a multimodal and multistage oncological approach that integrates individual brain plasticity profiling as a support to a step-by-step therapeutic decision making.

**Abstract:**

The ability of neural circuits to compensate for damage to the central nervous system is called postlesional plasticity. In diffuse low-grade gliomas (LGGs), a crosstalk between the brain and the tumor activates modulations of plasticity, as well as tumor proliferation and migration, by means of paracrine and electrical intercommunications. Such adaptative mechanisms have a major impact on the benefits and risks of oncological treatments but are still disregarded by current neuro-oncological guidelines. In this review, the authors first aimed to highlight clinical, radiological, and oncological markers that robustly reflect the plasticity potentials and limitations in LGG patients, including the location of the tumor and the degree of critical white matter tract infiltration, the velocity of tumor expansion, and the reactional changes of neuropsychological performances over time. Second, the interactions between the potential/limitations of cerebral plasticity and the efficacy/tolerance of treatment options (i.e., surgery, chemotherapy, and radiotherapy) are reviewed. Finally, a longitudinal and multimodal treatment approach accounting for the evolutive profiles of brain plasticity is proposed. Such an approach integrates personalized predictive models of plasticity potentials with a step-by-step therapeutic decision making and supports onco-functional balanced strategies in patients with LGG, with the ultimate aim of optimizing overall survival and quality of life.

## 1. Introduction

Supratentorial adult low-grade gliomas (LGGs) are a subgroup of histologically and molecularly defined diffuse primary brain neoplasms, including grade 2 IDH-mutant astrocytomas and grade 2 IDH-mutant and 1p19q co-deleted oligodendrogliomas [1]. Early stages of LGG expansion are characterized by a slow invasion of the brain parenchyma (generally, the mean diameter of expansion stands around 2–4 mm per year [2]) with a prominent tropism toward white matter microstructures [3,4]. Such diffusive pattern irremediably leads to an accumulation of genetic and epigenetic alterations that ultimately induce malignant transformation (MT) [5,6,7] and compromise patients’ neurological status. In parallel, as glioma cells are physically [8] and electrochemically integrated to the surrounding neural circuits [9,10], a bidirectional relationship exists between the tumor and its host brain: (i) the glioma proliferation is triggered by microstructural electrical [11] and regional functional activities [12], and is further affected by whole-brain network dynamics [13]; (ii) the functional connectivity between gliomas and other brain areas, and in-between remotely situated brain areas, is intensely reshaped to maintain network efficiency and cognitive performances (a compensatory mechanisms also called lesion-induced neuroplasticity, or plasticity) [14,15]. Such adaptative compensations remain constrained by the intrinsic anatomofunctional architecture of the brain [16,17] and do not have an infinite potential, since it may rapidly be overwhelmed in case of MT, thus affecting patients’ cognitive outcomes and clinical course [18,19]. Furthermore, the crosstalk between glioma growth and brain adaptation is perpetually evolving, resulting in significant constraints regarding the efficacy of oncological treatments and their tolerance in terms of cognitive maintenance and quality of life (QOL) preservation [20]. Reciprocally, each therapeutic step may have a decisive impact on short and long-terms plasticity modulations, since treatments may fragilize microcircuits that convey efficient functional remodeling. Therefore, the traditional oncological paradigm whereby the efficacy of one treatment is evaluated at one timepoint without an overall perspective of cumulative medical strategies and ongoing brain-to-glioma interactions can hardly be applied to LGGs. This is the reason why individualized and longitudinal oncological strategies have emerged over the last decade [21,22]: in these models, therapeutic options are considered in a holistic approach and traditional treatments can be postponed, repeated, or reversed, depending on individual parameters (e.g., tumor volume, effect of the previous line of treatment, functional status) with the ultimate goal of preserving the QOL in the long run. Strikingly, this strategy was shown to be effective, not only to expand the overall survival beyond 16 years in IDH-mutant LGGs, but also to sustain patients’ functional status in the long run (the Karnofsky performance status was maintained over 80% during 12.2 years in median in the Nancy experience [23]).

Yet, despite past and recent findings supporting the idea that plasticity is highly impactful on the clinical course of patients harboring LGGs, neuro-oncological guidelines do not currently implement neuroplasticity biomarkers in the decision-making processes that govern treatment strategies [24]. In this narrative review, the authors challenge this traditional approach by (i) highlighting the different clinical, radiological, and oncological markers associated with the efficiency of cerebral reorganizations (i.e., representative of plasticity potentials and limitations) in glioma patients, (ii) by emphasizing the critical interactions between the potential/limitations of cerebral plasticity and the efficacy/tolerance of oncological treatments, and (iii) by proposing a longitudinal approach that integrates the individual and evolutive profiles of brain plasticity to the multistage therapeutic decision making in LGG oncological strategies.

## 2. Treatment Goals in LGGs Clinical Management

### 2.1. Functional Goals

LGGs are generally diagnosed in young patients (on average aged from 30 to 40 years) [25,26] who enjoy normal social and professional lives. The overall survival of these patients has considerably evolved over the last two decades, mainly due to the systematization of extensive and early surgical approaches in comparison to “wait and watch” attitudes [27]. However, the natural history of the disease and the cumulative adverse effects of oncological therapies may have a drastic impact on patients’ QOL, including individual professional perspectives [28], wellbeing [29], and cognitive functioning, whether at early stages of the disease (and even before the first therapeutical steps [30]) or at the later stages of the oncological management [31]. Consequently, preserving QOL in the long run remains the main objective of the oncological strategies. In parallel to epileptic seizure control, which is the most common symptom in LGG patients [32], health-related QOL highly depends on a combination of social, behavioral, and neuropsychological factors [33]. Two main clinical endpoints have progressively emerged as landmark measures in the recent literature: (i) neurocognitive functioning, which should be evaluated with longitudinal neuropsychological assessments [34,35], and (ii) return to work [36], since the rate of resumption of professional activities reflects the ability of patients to maintain complex, integrated, goal-directed cognitive and behavioral tasks [37], while it also indicates the societal and economic impact of the treatments [38].

### 2.2. Oncological Goals

The aim of the oncological management of diffuse LGGs is to maintain permanent control over lesion size, in order to reduce the risk of MT. Although most assessments of therapeutic response in glioma trials are based on the response assessment in neuro-oncology (RANO) criteria (which are not directly based on lesion 3D volume but on bidirectional 2D measurements performed on a single slice), the 3D fluid attenuated inversion recovery (FLAIR) volume is by far the most effective radiological measure to assess objective treatment response, with the highest inter-reader agreement [39] and the highest levels of correlation with the risk of MT [40,41,42]. Importantly, both the pre-therapeutic volume [43] and the post-therapeutic volume have been recently revalidated as independent prognostic parameters across all histomolecular subclasses of LGGs [44], with an impact in terms of progression-free survival (postsurgical volume threshold <9.75 mL) and overall survival (postsurgical volume threshold <4.6 mL) [45]. Furthermore, although it is clear that every mL of tumor removed has a long-term impact on the course of the disease, the 10 ± 5 mL volume threshold was consistently established as a strong prognosis factor associated with MT and clinical outcomes [40,41,42,45,46]. Of note, current guidelines and ongoing trials do not account for this volume threshold and keep on employing the historical dichotomy between “low risk” and “high risk” LGG patients [47,48,49], with the age and a less-than-total gross resection as criteria to recommend early adjuvant therapies. Beyond the necessary need to integrate molecular mutations (e.g., IDH mutation, 1p19q codeletion, CDKN2A/B homozygous deletion), the arbitrary cutoff of 40 years and the lack of consideration of recent studies regarding the prognosis impact of the volumetric threshold reduce the scope of this dichotomy [50], and current oncological teams tend to defer adjuvant treatment in subcategories of “high risk” profiles [51], including in selected patients with grade 3/4 foci [52]. The aim of this attitude is to reduce the late adverse effects of oncological treatments (especially radiotherapy, see below).

### 2.3. Toward an Individual Onco-Functional Balance

Therefore, weighting the value of tumor cytoreduction versus neurological and neurocognitive worsening should be the main purpose when opting for a therapeutic option, not only with short term considerations, but also by anticipating long-term and cumulative adverse effects. The equilibrium between oncological benefits and neurocognitive risks defines the so-called “onco-functional balance” [53], which notably applies to intrasurgical decisions: the surgeon can voluntarily abandon, at that time, a tumor residual within cortical or white matter (WM) areas that subserve critical cognitive functioning to maintain QOL. However, this concept widely surpasses the intrasurgical decision making, since it may encourage sequential and delayed reoperation to optimize the extent of resection without inducing a cognitive deficit, given the fact that plasticity follows an evolving pattern [17]. It can also be applied to radiotherapy strategies, given the negative impact of WM irradiations on neurocognitive functioning [54], or even to chemotherapy, since some patients may not be eligible for sufficient surgical cytoreduction (typically when more than 10 ± 5 mL of tumor infiltrates functional WM tracts). In this setting, patients may benefit from a first line “neoadjuvant” chemotherapy before being reconsidered for a radical surgical (re)treatment [55]. These practical applications of the onco-functional balance urge neuro-oncologists to reconsider individual brain plasticity profiling as a cornerstone of oncological strategies, since brain compensation highly interferes with the benefits and risks of surgery and radiotherapy on neural structures.

In addition, individualized approaches should also integrate each patient’s wishes, professions, and cultural environment [56], which are the epicenters of modern, personalized “à la carte” oncological therapies. For instance, patients with specific professional activities at risk for legal or cultural reasons might benefit from different approaches (e.g., the case of an airplane pilot, free of any seizure, who could never fly again despite ensuring a complete resection after incidental discovery of a LGG was reported [57]). In the same vein, the whole onco-functional balance might be reconsidered in women followed for a LGG who project to have a baby, due to the major impact of pregnancy on tumor growth [58] and the critical effect of preventive total resections on overall survival in this subpopulation [59].

## 3. Profiling Individual Brain Plasticity in the Context of LGG

Neuroplasticity is the ability of neural networks to compensate the functional consequences of an injury to the brain architecture. Such ability has been observed in various pathological conditions, including traumatic brain injury [60], ischemic stroke [61,62], and brain tumors [15]. The mechanisms that govern cerebral remodeling in the event of LGG invasion remain poorly understood, but are thought to involve biochemical, paracrine, and synaptic transmission in parallel to macroscale network remodeling [18], including local and brain-wide modulations of functional brain activities [63,64] and volume inflations within associated cortical areas [65,66,67]. Several clinical and radiological measures have been highlighted as conclusive biomarkers of neuroplasticity (Figure 1) [20].

### 3.1. The Location of the Tumor within Brain Architecture and the Level of Infiltration within Critical WM Tracts

Anatomical structures infiltrated by the tumor show a variable propensity for functional reallocation. A group of WM pathways and unimodal cortices composes the “minimal connectome”, a common structure that displays a very low inter-individual variability and a minimal potential for plastic compensation [16]. Several studies based on functional-guided brain excision and intrasurgical electrostimulation mapping in LGG patients have provided a comprehensive atlas of human brain plasticity [68,69,70]. Overall, unimodal cortical areas and WM structures (especially associative WM tracts) were found to represent a major limitation of plastic potential, with low plastic indices [68]. Consequently, delineating the pattern of tumor diffusion within critical WM tracts is a necessary step to predict the expected extent of resection in a surgical perspective and the expected neurocognitive declines following surgery or radiotherapy.

### 3.2. The Velocity of Tumor Expansion

The velocity of the tumor, i.e., the slope of the mean tumor diameter growth curve extrapolated from consecutive volumetric measurements [71], is correlated with long-term clinical outcomes [72]. In addition, the kinetics of tumor expansion has a drastic impact on adaptative plastic mechanisms [73]. This is the reason why acute neurological lesions (e.g., stroke or brain trauma) or rapidly evolving brain tumors (e.g., glioblastomas or metastases) have typically a lower rate of functional compensation. Likewise, accelerations of the tumor expansion, especially at tumor recurrence, may encourage clinicians to select a preventive strategy by urging a local treatment in case of focal proliferative pattern, whether associated with the onset of contrast enhancements or not (i.e., when the diffusion toward WM remains restricted), versus by favoring systemic treatments such as chemotherapy in case of rapid anisotropic diffusion toward WM tracts. On the other hand, patients with so-called “high risk” profiles (i.e., age > 40 years or subtotal resection) with tumor residual <10 mL and low pre-therapeutic velocities of expansion should be considered for simple follow-up and vigilant MRI surveillance, since the low velocity of tumor expansion may favor evolving plasticity potentials and effective early reoperation strategies.

### 3.3. Cognitive Compensation

Neuropsychological assessments provide objective measures of plasticity efficiency. Such examinations should be repeated in a longitudinal manner before/after each therapeutical step. Cognitive declines suggest a high degree of functional alterations within critical circuits that cannot be compensated anymore [74]. Thus, it indicates a saturation of neuroplasticity mechanisms and a high risk of immediate cognitive decompensation in case of local surgical approach.

The time-dependent and non-linear characteristics of LGG-induced plasticity are the results of evolving patterns of neural activity and reallocations. Such modulations, integrated in a dynamic onco-functional framework, are named meta-plasticity [75,76]. Therefore, the three mentioned biomarkers (3-dimensional WM infiltration or “3D”, the velocity of expansion or “t(ime)”, and the meta-plastic cognitive compensation “m”, also referred as the “3Dtm” model [20]) are constantly evolving, depending on spontaneous and therapeutic events, and should be assessed at each stage of the disease to guide step-by-step, personalized therapeutic approaches.

## 4. The Plasticity Potential May Interfere with the Benefits and Risk of Treatment Options

### 4.1. Surgery

Surgical excision has become the treatment of choice of LGGs at diagnosis, given the results of several pseudo-randomized studies advocating for a clear survival advantage in patients undergoing surgery versus biopsy plus radio-chemotherapy [27,77]. In addition, language monitoring under awake condition [14] and intraoperative stimulation mapping techniques have been demonstrated as efficient methods to preserve motor and language functions (with a rate of permanent neurological deficit approaching 3% [78]), while allowing an increase in the extent of excision [79]. The evolution of cortical and subcortical mapping techniques towards integrated, multimodal awake cognitive monitoring [80] now allows the preservation of higher-level cognitive processes, beyond basic language and motor skills [81]. This paradigm has now been revalidated and applied to tumor excisions located in the right hemisphere, especially in regions historically described as “non eloquent” [82]. In parallel, the prognosis effect of the extent of resection on overall survival is widely accepted across all histomolecular subclasses [45], and there is now cumulative evidence that resecting the tumor beyond the abnormalities of the FLAIR signal (i.e., “supratotal” resection) significantly improves the long-term course of the disease [83,84].

Nonetheless, total or supratotal resections cannot be achieved in all patients, due to obvious limitations of plasticity potential [16]. Thus, assessing neuroplasticity potential via appropriate and routine clinic-radiological measures (i.e., degree of WM infiltration, velocity of expansion, cognitive compensation) is decisive to select eligible patients for extensive resections and to estimate the functional consequences regarding reasonable oncological goals (e.g., reducing the tumor volume under the 10 *±* 5 mL threshold, as illustrated in Figure 2A). Moreover, various plasticity atlases obtained from intraoperative electrostimulation findings [70,85] and functional resections [68] may help clinicians to guide presurgical planning and appreciate the probabilistic degree of functional compensation depending on the WM infiltration. A notable impact of neuroplasticity potential on surgical resectability is that earlier management generally results in smaller tumor volumes, less infiltration within the critical cortical and WM tracts, and better cognitive compensation. This is the reason why incidental LGGs usually benefit from higher rates of supratotal resections, better oncological prognosis [43], higher rates of cognitive preservation [86], and resumption of professional activities after preventive surgery [59]. The impact of such preventive management on the onco-functional framework is displayed in Figure 2B.

Importantly, LGG-induced plasticity potential is constantly evolving [17]. At the early stage of the disease, broadcasted schemes of functional hyperconnectivity were observed and may represent predictive markers of cognitive resilience to surgical excision [64]. Subsequently, cortical reorganizations may continue and open the door to resections of brain areas that were non-compensable during the first surgery and that were finally reallocated years or decades later [17,87,88]. Therefore, neuroplasticity markers should be assessed longitudinally, with repetitive appreciations of WM infiltration, velocity of expansion, and serial neuropsychological examinations. Patients with high plastic potential and whose tumor volume can still be reduced under the 10 mL threshold should benefit from reoperation (see an example in Figure 2A), given the excellent neurological and neuropsychological tolerance of iterative awake-guided resections [89,90] and the long-term benefits in terms of overall survival [90,91,92,93,94,95], even when the tumor invades crossroads and challenging anatomical regions such as the insula [96]. Further, some authors have demonstrated that functional-based surgery remains applicable for a third operation, years after the initial diagnosis, with excellent functional outcomes and a median overall survival reaching 17.8 years [97] (here illustrated in Figure 2C).

### 4.2. Chemotherapy

Procarbazine, CCNU, and vincristine (PCV) is an effective chemotherapy combination that was used for the first time in LGGs at the end of the 1990s [98,99,100]. Since then, the only significant addition to the medical armamentarium was temozolomide. Given temozolomide’s efficacy as an adjunct treatment in monotherapy (there was no difference of treatment efficacy between radiotherapy alone and temozolomide alone in “high risk” LGGs [47]), temozolomide was finally adopted by a vast majority of oncological centers, because of its better tolerance profile in comparison to PCV [51].

Of note, there are still few studies investigating the effect of chemotherapy on cerebral plasticity and vice versa. Although detrimental effects of temozolomide have been reported on hippocampal neurogenesis and theta activity in the cerebrum of rats [101], there are no clear reports documenting independent cognitive deteriorations related to temozolomide in humans [55]. On the other hand, in the event of intractable tumor-related seizures, chemotherapy may help to obtain seizure control and indirectly to promote further plasticity compensations [102]. Likewise, early chemotherapy may facilitate neuroplasticity potential by favoring a reduction of WM infiltration (see an illustration in Figure 2A,C), which can be evidenced by serial diffusion tensor imaging and functional diffusion maps [103]. Hence, introducing chemotherapy as a first-line treatment in patients with unfavorable neuroplastic markers (especially in tumor with highly diffuse patterns toward critical WM tracts) may open the door to radical excisions with higher probability of total/supratotal resections and excellent functional outcomes [55,104,105]. Further, this strategy may be repeated in several sessions over years to allow a long-term control of the disease [23].

Furthermore, beyond the dogma that recommend systematic early chemotherapy in so-called “high risk” glioma, postponing treatments in a subgroup of patients with residual <10 mL and favorable neuroplastic indices (especially in 1p19q oligodendroglioma) may help to delay the side effects of adjuvant treatments and to preserve therapeutic options at later stages of the disease [106]. Such a strategy may be justified by the potential long-term acquired resistances to temozolomide driven by mismatch repair genes [107], given the potential clinical impact of hypermutation profiles on the latest stages of glioma progression [108].

### 4.3. Radiotherapy

Radiotherapy is another cornerstone of oncological therapies in LGGs, with a well-demonstrated cytoreductive and anticonvulsant effect. Early uses of radiotherapy following surgical excision have shown benefits on progression-free survival, but no significant effects regarding overall survival [109]. However, the deleterious effect of radiation on cognition (particularly in terms of attentional and executive processes) has been reported in patients with long-term follow-up (>12 years), even at doses traditionally considered safe (<2 Gy) [110]. A recent report also advocates for earlier detrimental consequences of radiation, with up to 50% of patients suffering from cognitive and emotional disturbances after a median follow-up of 5 years [54]. The effects of modern radiation technologies (including intensity modulated radiotherapy, volumetric modulated arc therapy, or proton therapy) on long-term cognitive functioning remain unknown, with reports only reaching a limited follow-up [111,112] and usually based on restrained neuropsychological batteries (e.g., mini-mental state examination) [112,113]. Over the past 15 years, significant changes have been observed regarding the use of radiotherapy in IDH-mutant LGGs [23,51]. Most teams seem to postpone the introduction of radiotherapy at later stages of the disease, especially in 1p19q codeleted LGGs [114] and MGMT-methylated LGGs [115], to reduce the neurocognitive effects of radiotherapy in patients now surviving 16 years or more in median [23,45,106].

Despite significant technical advances and the use of proton therapy, which allows radiation exposure to surrounding brain structures to be minimized [116], two conceptual limitations tend to constrain the use of radiation therapy:(1)with the advent of functional-guided surgical techniques, residual tumor volumes targeted by radiation therapies inevitably consist in highly functional areas with restricted potential for plastic compensation [66], especially within the WM pathways forming the “minimal common brain” [16], which is essential to maintaining long-term cognitive compensation.(2)there are now sufficient evidence that radiotherapy leads to alterations of the WM microstructure through demyelination and axonal degeneration mechanisms [117], which can easily be measured with modern neuroimaging tools (e.g., diffusion tensor imaging) [118,119]. The resulting structural and functional connectivity alterations may be correlated with early [120] and long-term [121] cognitive performance, although brain regional susceptibilities are still poorly documented [122].

Overall, radiotherapy strategies have remained dissociated from current knowledge on LGG-induced neuroplastic potential and its main limitations (in particular, critical associative WM bundles). The study of the effects of radiotherapy on WM microstructure alterations and their translation into cognitive and motor deficits represent an expanding field of neuro-oncology research [123,124]. Beyond the preservation of essential cortical structures such as the hippocampus [125,126], it is now essential that future guidelines integrate current knowledge on LGG-induced plasticity and functional data obtained from awake surgery to adapt treatment planification. Modulations of therapeutic targets and multistage irradiations accounting for histomolecular data (e.g., MGMT status, 1p19q codeletion) and intrasurgical findings (critical functional WM tracts) may help to optimize the onco-functional balance and to delay cognitive declines [127].

### 4.4. Other Therapies

Several treatment options are currently under investigation. Among them, multiple strategies of vaccine therapy are under development and seemed to be associated with promising intratumoral inflammatory reactions [128].

However, the most advanced candidates are represented by therapies targeting IDH mutations [129]. In particular, vorasidenib, a dual inhibitor of IDH1/2 mutation, was shown as a well-tolerated therapy with conclusive brain penetrance in patients with recurrent gliomas [130] and significant effects on progression-free survival after a 14-month follow-up in patients who only benefited from previous surgical approaches [131]. However, the long-term efficacy of anti-IDH therapies, including their effect on MT, their cumulative cognitive tolerance (as a sole treatment and in combination with other chemotherapies), and their long-term effects on the glioma mutational charges, are required before integrating such molecules into the landscape of therapeutic options against LGGs. Beyond the stabilization of the FLAIR signal, a critical pending question resides in the ability of IDH inhibitors to reduce WMT infiltration (as observed after temozolomide administration). Such data are required to inform the optimum timeline of IDH inhibitor administration in comparison to temozolomide, especially in patients who can benefit from a neoadjuvant systemic approach before being operated upon.

## 5. Proposal of a Multistage and Individualized Workflow Accounting for the Plasticity Potential

Here, our aim was to propose an integrative framework based on personalized, multistage, and longitudinal therapeutic approaches, with step-by-step adaptations of therapeutic options, depending on oncological features on the one hand (e.g., tumor volume, histomolecular feature) and the evolution of clinic-radiological markers of neuroplasticity on the other hand (Figure 3). Similar approaches have been previously developed and successfully validated [21,23] in large populations of LGGs. Here, it has been revised to dynamically accounts for markers of plasticity according to the “3Dtm” model [20]. It challenges the standard therapeutic attitude by proposing earlier and preventive radical treatment, repetitive surgical treatment in patients with favorable plasticity markers, and possibly a reversal of the classical order of treatments. The goal is primarily to maintain QOL in the long run, while maintaining control over tumor volume and reducing the risk of MT in a dynamic framework integrating the constantly evolving plasticity potential (i.e., meta-plasticity) that results from the brain-to-glioma crosstalk. In addition, by postponing and saving therapeutic options, this oncological philosophy allows for anticipating medical strategies several steps ahead, with an aim to prevent clinicians from running out of treatment resources at the latest stages of the disease.

Prospective studies with long-term follow-up are required to replicate current findings and improve the level of evidence of the proposed management. Because randomized controlled trials are definitely not the best method to investigate such a global and adaptative strategy, the authors suggest that significant efforts should be made to build/improve national and multicenter registries, which could help to measure the effects of different combinations of treatment with sufficient follow-up, as long as systematic records include clinical (neurological and neuropsychological status, working abilities), radiological (volumes, velocity of expansion), and histomolecular outcomes.

## 6. Conclusions and Perspectives

The constant interplay between gliomas and the central nervous system is a perpetually evolving neurobiological process that considerably modulates the impact of current therapeutic resources deployed against LGGs, the overall oncological prognosis and the cognitive resilience of patients throughout the history of their disease. Strikingly, it has been neglected by practical oncological guidelines. Thus, the first step to improve the efficiency of the current armamentarium would be to reintegrate objective radio-clinical markers of brain compensation into the decision-making strategies, in order to improve the long-term oncological and functional prognosis of patients who will benefit from numerous sequential, combined, and repeated therapeutic regimens throughout their oncological course. Beyond controlling the volume of the tumor and delaying MT, future game-changing therapies should be developed with the main goal of reducing the migration of LGG cells toward axonal connectivity, which considerably restrains the plasticity potential. A second perspective would be to improve the identification of plasticity mechanisms, by means of multimodal objective neuroimaging tools (e.g., task positive or resting state functional MRI) and/or non-invasive brain mapping of cognitive compensation (e.g., by mean of navigated transcranial stimulation [132]). Such prospects could help clinicians to optimize the timeline of treatment selection. A third perspective would be to potentiate neuroplastic reorganizations by surpassing the functional limits represented by the minimal connectome. To achieve this goal, further studies investigating the mechanisms that underly circuit remodeling (especially functional network reshaping toward distant ipsilateral or contralateral regions and the dynamics of meta-network reconfigurations [133]) are definitely required to advise potential neuromodulation protocols (e.g., repetitive transcranial stimulation or transcranial direct current stimulation [134]).

## Figures and Tables

**Figure 1 cancers-15-03698-f001:**
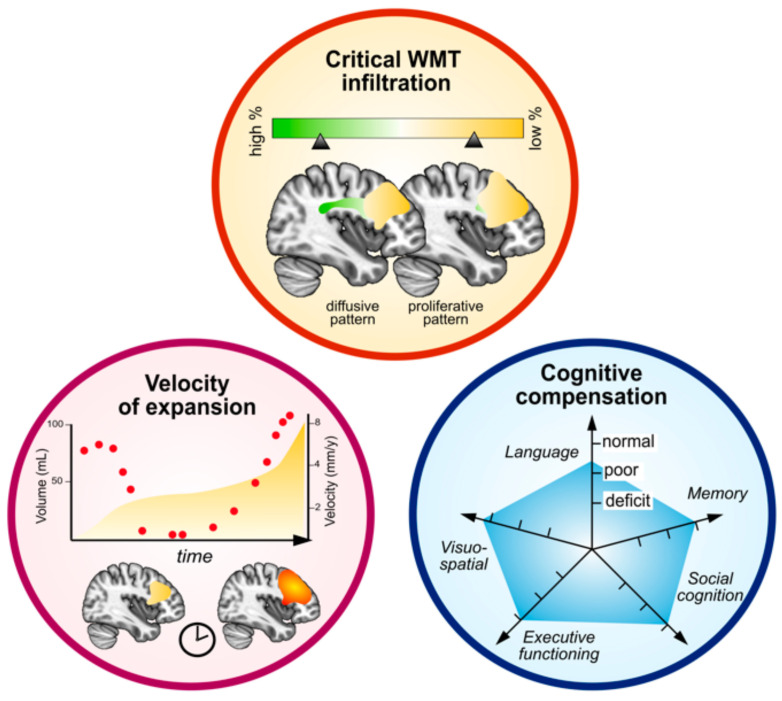
Biomarkers of plasticity potential in patients harboring low-grade gliomas. These main clinic-radiological markers refer to the “3Dtm” model from Duffau [20] (3D: tumor location and white matter infiltration; t: velocity of expansion over time; m: metaplasticity compensation, clinically assessed with repeated cognitive evaluations). WMT: white matter tracts.

**Figure 2 cancers-15-03698-f002:**
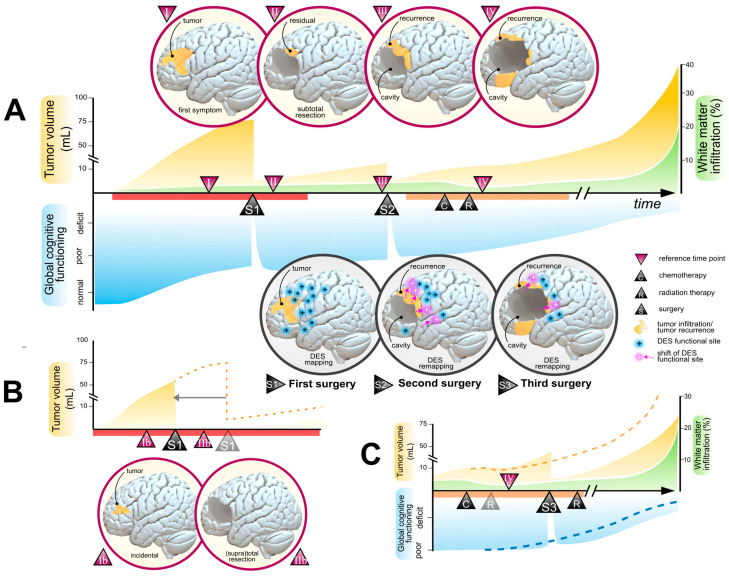
Global and dynamic onco-functional framework in an illustrative case of low-grade glioma, adapted from Ng et al. [76]. (**A**) Global onco-functional framework in a patient with a symptomatic 1p19q non-codeleted IDH-mutant low-grade glioma located in the left frontal lobe (reference time point I) who underwent a subtotal resection assisted with awake mapping (S1). The reference time point II shows tumor residual with a volume <10 mL. Given the favorable individual plasticity profile at recurrence (reference time point III) with a limited degree of white matter tract infiltration, no modification of the velocity of tumor expansion, and a preserved cognitive status, the patient was proposed a second surgery assisted with awake mapping (S2). Plasticity mechanisms promoted significant reallocations of functional sites, thus allowing the reduction of the volume under 10 mL again. The patient was finally administered chemotherapy (C) and radiotherapy (R) several years after the first line of oncological treatment (reference time point IV). (**B**) Early dynamic adaptation of the onco-functional strategy in a similar patient who was incidentally diagnosed (reference time point Ib). The patient was preventively operated with awake mapping (S1) and benefited from a supratotal resection (reference time point IIb), with regular surveillance but without adjuvant treatment. (**C**) Late dynamic adaptation of the onco-functional strategy: the patient underwent a third surgery (S3) at the reference time point IVc instead of radiotherapy, since chemotherapy significantly helped to reduce the white matter tract infiltration—while the glioma was an astrocytoma IDH1 mutated but 1p19q non codeleted. Evolving patterns of plasticity again allowed a subtotal resection with a tumor residual <10 mL. DES: direct electrostimulation. Note that transient cognitive declines following surgeries are common after awake, functional-guided surgery, due to brain edema (here illustrated with the blue curve inflexion following each surgical procedure). Note that the linearity of the time scale is not necessarily respected in these illustrations.

**Figure 3 cancers-15-03698-f003:**
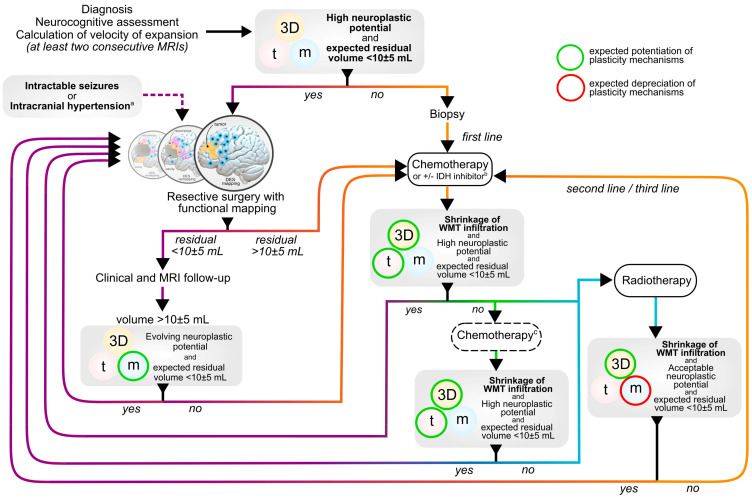
Proposal of a dynamic therapeutic strategy accounting for individual brain plasticity profiling (3Dtm model), adapted from Duffau and Taillandier [21]. “3D” refers to the location of the tumor and the degree of critical white matter tract infiltration (see Ius et al. [16], Herbet et al. [66] and Sarubbo et al. [68] for dedicated atlas of functional plasticity); “t” refers to the velocity of tumor expansion; “m” refers to the longitudinal cognitive compensation (as a result of meta-plasticity, which also promote evolutive functional reallocations, as reported by Ng et al. [17]). ^a^ resective surgery with functional mapping should be considered at each stage of the disease in case of intractable seizures or in case of intracranial hypertension. ^b^ IDH inhibitors show promising results if administered before a first line of chemotherapy but are not currently validated regarding the absence of safety and efficacy results at mid- and long-term follow-up. ^c^ in case of IDH inhibitor use as a first line of systemic treatment, otherwise radiotherapy should be considered. DES: direct electrostimulation. MRI: magnetic resonance imaging.

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
