# Peer review of "Brain Plasticity Profiling as a Key Support to Therapeutic Decision-Making in Low-Grade Glioma Oncological Strategies"

_cancers, 2023, doi:10.3390/cancers15143698_

Round 1

Reviewer 1 Report

The argument proposed by the authors is interesting; nevertheless, the literature landscape already presents similar articles, some of which published by the authors (Ref. 21-22). In my opinion, to guarantee an innovative manuscript the authors should modify this draft in a systematic review according to PRISMA guidelines; the discussion and future perspectives could costitute the strong conclusion of the available literature.

Furthermore, other pitfalls should be clarified and eventually corrected by the authors:

- The Title is unclear; the main aim should be understood in the first sentences.

- The authors should quote the studies regarding the mathematical models in gliomagenesis (I suggest the group of Acerbi and Ciarletta - Milan).

- Please define the LGG in the study as "adult LGG", since pediatric histotypes are classified separately. (Line 130-131): define the grading in arabic numbers.

- how about fluorophores as intraoperative adjunctive tools? Which is the role of functional and DTI preoperative MRI as well as modern software (i.e., Quicktome).

After a wide process of revision, I think that this manuscript could be accepted for publication.

English sounds good with a favorable syntaxis; minimal linguistic choices could be ameliorated.

Author Response

We thank the referee #1 for her/his comment.

1) Regarding the interest of a PRISMA review, the referee should be informed that the current manuscript is an invited narrative review, which intentionally does not correspond to the format of a systematic review.

2) We respectfully disagree with the comment of the referee regarding the Title of our manuscript. In agreement with the comments of the other three referees, we decided to maintain our original title.

3) To the authors’ knowledge, there are no recent works dealing with gliomagenesis in low-grade gliomas from the group of Acerbi and Ciarletta (Milano). Recent studies from this group investigate biomathematical modelling in IDH wild-type glioblastomas and were not found relevant to discuss low-grade glioma biogenesis and plasticity-related mechanisms (see references hereafter).

1- A Mechanical Model for Glioblastoma Multiforme Growth including Brain Hyperelasticity and Patient-Specific Data, Lucci et al, Proceedings of Simai 2021

2- In silico mathematical modelling for glioblastoma: a critical review and a patient-specific case, Falco et al, Journal of Clinical Medicine 2021

3- Mechano-biological features in a patient-specific computational model of glioblastoma, Acerbi et al, Brain Tumors 2020

4- Learning patientspecific parameters for a diffuse interface glioblastoma model from neuroimaging data, Agosti et al, Mathematical methods and applied Sciences 2020

5- A computational framework for the personalized clinical treatment of glioblastoma multiforme, Agosti et al, ZAMM 2018

6- Towards the personalized treatment of glioblastoma: integrating patient-specific clinical data in a continuous mechanical model, Colombo et al PloS one 2015

4) Regarding pediatric low-grade gliomas, we added the “adult” mention to the first line of our introduction:

“Supratentorial adult low grade gliomas (LGGs) are a subgroup of histologically and molecularly defined diffuse primary brain neoplasms, including grade 2 IDH-mutant astrocytomas and grade 2 IDH-mutant and 1p19q co-deleted oligodendrogliomas” line 39-41

In accordance with the comment of the reviewer, all mentions of histopathological grades are now stated in Arabic numbers (as recommended by the WHO 2021 classification):

“current oncological teams tend to defer adjuvant treatment in subcategories of “high risk” profiles [51], including in selected patients with grade 3/4 foci” line 129-131

5) Regarding intraoperative adjunctive tools: note that neither fluorophores nor quicktome tool are validated for the surgical approach of low-grade gliomas. Intraoperative stimulation mapping is currently the gold standard approach (see De Witt Hammer et al, JCO 2012 and Hervey Jumper et al, JCO 2023).

Reviewer 2 Report

This is a very interesting, stimulating and well-argued review. The paper deserves to be published in Cancer, since neuro-oncologists can learn about the evolution of low-grade gliomas but also connect with a clinical management proposal that can take advantage of the brain plasticity. The text is well written and the figures are informative. However, the proposed management is not based on empirical results ,which is understandable, but the authors do not discuss this issue or propose a methodology to provide more evidence for their approach. 

IDH inhibitors may change the paradigm of LGG treatment and these drugs may delay or avoid radiotherapy, in line with the current proposal, but the authors hardly comment on the benefits of IDH inhibitors. 

In the light of these comments, I would recommend two minor additions to the manuscript:

-A comment on the possibility of a study that could improve the evidence for the proposed management. 

-A comment on the possibilities of IDH inhibitors to allow brain plasticity in LGG. 

Author Response

We thank the referee #2 for her/his manifested interest in our work.

1) Regarding potential studies that could improve the evidence for the proposed management:

Prospective studies with long-term follow-up are required to replicate current findings and improve the level of evidence of the proposed management. Because randomized controlled trial are definitely not the good method to investigate such a global and adaptative strategy, the authors suggest that significant efforts should be made to build/improve national and multicenter registries, which one could help to measure the effect of different combinations of treatment with sufficient follow-up, as long as systematic records include both clinical (neurological and neuropsychological status, working abilities), radiological (volumes, velocity of expansion) and histomolecular outcomes.

The following lines were added to the revised manuscript:

“Prospective studies with long-term follow-up are required to replicate current findings and improve the level of evidence of the proposed management. Because randomized controlled trial are definitely not the good method to investigate such a global and adaptative strategy, the authors suggest that significant efforts should be made to build/improve national and multicenter registries, which one could help to measure the effect of different combinations of treatment with sufficient follow-up, as long as systematic records include both clinical (neurological and neuropsychological status, working abilities), radiological (volumes, velocity of expansion) and histomolecular outcomes.” Lines 411-418

2) Regarding the possible contribution of IDH inhibitors:

The effect of IDH inhibitors on neurobiological interactions between the brain and the tumor is unknown to date. Further data on efficacy (based on primary endpoints different from the progression free survival established on RANO criteria, and different from time to the next intervention, which is inconstant in terms of indication from one neuro-oncological team to another) and safety (by means of neurocognitive measures) with longer follow-up are necessary. Beyond the stabilization of the FLAIR signal, a critical pending question resides in the ability of IDH inhibitors to reduce WMT infiltration (as observed after temozolomide administration). Such data are required to inform the optimum timeline of IDH inhibitor administration in comparison to temozolomide, especially in patient who can benefit from a neoadjuvant systemic approach before being operated.

The following lines were added to the revised manuscript:

“Beyond the stabilization of the FLAIR signal, a critical pending question resides in the ability of IDH inhibitors to reduce WMT infiltration (as observed after temozolomide administration). Such data are required to inform the optimum timeline of IDH inhibitor administration in comparison to temozolomide, especially in patient who can benefit from a neoadjuvant systemic approach before being operated.” Lines 387-392

Reviewer 3 Report

In their review, Sam Ng and Hugues Duffau provide a critical analysis on the therapeutical strategies used to treat low-grade glioma.

The review encompasses all aspects related to the therapeutic management of patients with low-grade glioma. It summarizes the different elements taken into account when choosing the most appropriate therapeutic option: the characteristics of the tumor and its evolution, and the patient's capacity for cognitive compensation. The impact of this plasticity potential on the benefits and risks of therapeutic options is then assessed.

The review is well written, easy to understand for scientists and surgeons. There are a few areas for improvement:

1-      Line 203 so-called high risk profiles, it would be helpful to the reader if the authors could define what is meant by "high risk".

2-      With regard to plasticity, the authors focused on functional plasticity, which is the most important for individuals, but effects on neuroanatomical changes were also reported, such as changes in cortical volumes. Below are a few examples of publications related to this point:

Cortex. 2022 Dec;157:245-255.doi: 10.1016/j.cortex.2022.09.014. Language reorganization in patients with left-hemispheric gliomas is associated with increased cortical volume in language-related areas and in the default mode network

Aging (Albany NY). 2020 Jun 5;12(11):10259-10274. doi: 10.18632/aging.103212. Structural plasticity of the bilateral hippocampus in glioma patients

BMJ Case Rep. 2019 May 5;12(5):e228971.doi: 10.1136/bcr-2018-228971. Spatial reorganisation of the somatosensory cortex in a patient with a low-grade glioma

3-      As far as radiotherapy is concerned, it would be interesting to mention the development of techniques designed to minimize the long-term sequelae of radiotherapy. (for example: Dosimetric advantages of proton therapy over conventional radiotherapy with photons in young patients and adults with low-grade glioma., Strahlenther Onkol, 2016 Nov;192(11):759-769, doi: 10.1007/s00066-016-1005-9.)

Author Response

We thank the referee #3 for her/his manifested constructive comment.

1) We added the following lines:

“On the other hand, patients with so-called “high risk” profiles (i.e. age>40 years or subtotal resection) with tumor residual <10mL and low pre-therapeutic velocities of expansion should be considered for simple follow-up and vigilant MRI surveillance.” Line 203

2) Regarding plasticity-induced structural changes, the following additions were made to the revised manuscript:

“The mechanisms that govern cerebral remodeling in the event of LGG invasion remain poorly understood but it is thought to involve biochemical, paracrine, and synaptic transmission in parallel to macroscale network remodeling,[18] including local and brain-wide modulations of functional brain activities [63,64] and volume inflations within associated cortical areas [65-67].” Lines 169-173

  1. Almairac, F.; Duffau, H.; Herbet, G. Contralesional Macrostructural Plasticity of the Insular Cortex in Patients with Glioma: A VBM Study. Neurology 2018, 91, e1902–e1908, doi:10.1212/WNL.0000000000006517.
  2. Yuan, T.; Ying, J.; Zuo, Z.; Gui, S.; Gao, Z.; Li, G.; Zhang, Y.; Li, C. Structural Plasticity of the Bilateral Hippocampus in Glioma Patients. Aging 2020, 12, 10259–10274, doi:10.18632/aging.103212.

67.Pasquini, L.; Jenabi, M.; Peck, K.K.; Holodny, A.I. Language Reorganization in Patients with Left-Hemispheric gliomas Is Associated with Increased Cortical Volume in Language-Related Areas and in the Default Mode Network. Cortex 2022, 157, 245–255, doi:10.1016/j.cortex.2022.09.014.

3) Regarding proton therapy, the reference recommended by the referee was added to the manuscript:

“Despite significant technical advances and the use of proton therapy, that allows to minimize radiation exposure to surrounding brain structures [116]” Lines 349-351

  1. Harrabi, S.B.; Bougatf, N.; Mohr, A.; Haberer, T.; Herfarth, K.; Combs, S.E.; Debus, J.; Adeberg, S. Dosimetric Advantages of Proton Therapy over Conventional Radiotherapy with Photons in Young Patients and Adults with Low-Grade Glioma. Strahlenther Onkol 2016, 192, 759–769, doi:10.1007/s00066-016-1005-9.

Reviewer 4 Report

I have carefully reviewed your manuscript titled "Brain plasticity profiling as a key support to therapeutic decision-making in low-grade glioma oncological strategies." I would like to commend you on providing valuable insights into the role of postlesional plasticity in diffuse low-grade gliomas (LGGs) and its implications for oncological treatments. Overall, I find the manuscript to be well-written and comprehensive in its coverage of the subject matter.

I appreciate the thorough identification and discussion of various clinical, radiological, and oncological markers that reflect the plasticity potentials and limitations in LGG patients. The examination of the interactions between cerebral plasticity and treatment options such as surgery, chemotherapy, and radiotherapy is particularly insightful. Your proposed longitudinal and multimodal treatment approach, which integrates personalized predictive models of plasticity potentials, shows promise in optimizing overall survival and quality of life for LGG patients.

Considering the significance of your findings, I believe this manuscript will be of great interest to researchers and clinicians in the field of glioma and medical sciences. I have no further comments or suggestions regarding the content or structure of the manuscript.

The manuscript was well-written and the quality of English language seems fine.

Author Response

We thank the Referee#4 for her/his encouraging comment.

Round 2

Reviewer 1 Report

\